# A population-based study exploring phenotypic clusters and clinical outcomes in stroke using unsupervised machine learning approach

Ralph K. Akyea[1☯‡*], George Ntaios[2], Evangelos Kontopantelis[3,4], Georgios Georgiopoulos[5], Daniele Soria[6], Folkert W. Asselbergs[7,8], Joe Kai[1], Stephen F. Weng[1☯‡], Nadeem Qureshi[1☯‡]

1 PRISM Research Group, Centre for Academic Primary Care, School of Medicine, University of Nottingham, Nottingham, United Kingdom, 2 Department of Internal Medicine, Faculty of Medicine, School of Health Sciences, University of Thessaly, Larissa, Greece, 3 Division of Population Health, Health Services Research and Primary Care, School of Health Sciences, Faculty of Biology, Medicine and Health, Manchester Academic Health Science Centre (MAHSC), The University of Manchester, Manchester, United Kingdom, 4 Division of Informatics, Imaging and Data Sciences, School of Health Sciences, Faculty of Biology, Medicine and Health, Manchester Academic Health Science Centre (MAHSC), The University of Manchester, Manchester, United Kingdom, 5 School of Biomedical Engineering and Imaging Sciences, St Thomas Hospital, King's College London, London, United Kingdom, 6 School of Computing, University of Kent, Canterbury, United Kingdom, 7 Amsterdam University Medical Centers, Department of Cardiology, University of Amsterdam, Amsterdam, The Netherlands, 8 Health Data Research UK and Institute of Health Informatics, University College London, London, United Kingdom

☯ These authors contributed equally to this work.
‡ These authors are joint senior authors on this work.
* Ralph.Akyea1@nottingham.ac.uk

**Data Availability Statement:** Restrictions apply to the availability of these data, which were used

## Abstract

Individuals developing stroke have varying clinical characteristics, demographic, and biochemical profiles. This heterogeneity in phenotypic characteristics can impact on cardiovascular disease (CVD) morbidity and mortality outcomes. This study uses a novel clustering approach to stratify individuals with incident stroke into phenotypic clusters and evaluates the differential burden of recurrent stroke and other cardiovascular outcomes. We used linked clinical data from primary care, hospitalisations, and death records in the UK. A data-driven clustering analysis (kamila algorithm) was used in 48,114 patients aged ≥ 18 years with incident stroke, from 1-Jan-1998 to 31-Dec-2017 and no prior history of serious vascular events. Cox proportional hazards regression was used to estimate hazard ratios (HRs) for subsequent adverse outcomes, for each of the generated clusters. Adverse outcomes included coronary heart disease (CHD), recurrent stroke, peripheral vascular disease (PVD), heart failure, CVD-related and all-cause mortality. Four distinct phenotypes with varying underlying clinical characteristics were identified in patients with incident stroke. Compared with cluster 1 (n = 5,201, 10.8%), the risk of composite recurrent stroke and CVD-related mortality was higher in the other 3 clusters (cluster 2 [n = 18,655, 38.8%]: hazard ratio [HR], 1.07; 95% CI, 1.02–1.12; cluster 3 [n = 10,244, 21.3%]: HR, 1.20; 95% CI, 1.14–1.26; and cluster 4 [n = 14,014, 29.1%]: HR, 1.44; 95% CI: 1.37–1.50). Similar trends in risk were observed for composite recurrent stroke and all-cause mortality outcome, and

under license for the current study, and so are not publicly available. The data that support the findings of this study are available from Clinical Practice Research Datalink (CPRD) through a data request application process (https://cprd.com/data-access). Researchers can contact enquiries@cprd.com for more information.

**Funding:** RKA was funded by a National Institute for Health Research School for Primary Care Research (NIHR SPCR) PhD Studentship award, supervised by NQ, FWA, and JK. The funders had no role in study design, data collection and analysis, decision to publish, or preparation of the manuscript.

**Competing interests:** I have read the journal's policy and the authors of this manuscript have the following competing interests: SFW has received independent research grant funding from AMGEN. NQ and SFW have previously received honorarium from AMGEN. RKA currently holds an NIHR-SPCR funded studentship (2018-2021). SFW is currently an employee of GSK. FWA is supported by UCL Hospitals NIHR Biomedical Research Centre. The remaining authors have no competing interests.

subsequent recurrent stroke outcome. However, results were not consistent for subsequent risk in CHD, PVD, heart failure, CVD-related mortality, and all-cause mortality. In this proof of principle study, we demonstrated how a heterogenous population of patients with incident stroke can be stratified into four relatively homogenous phenotypes with differential risk of recurrent and major cardiovascular outcomes. This offers an opportunity to revisit the stratification of care for patients with incident stroke to improve patient outcomes.

## Author summary

Using an unsupervised machine learning cluster analysis approach, adult patients with incident stroke were grouped into four clinically meaningful phenotypic clusters based on their demographic, biochemical, comorbidities, and prescribed medication profiles at the time of incident stroke. The findings of this study highlight the significant heterogeneity that exists within patients with incident stroke with respect to subsequent cardiovascular morbidity and mortality outcomes. This offers an opportunity to revisit the stratification of care for patients with incident stroke to improve patient outcomes and highlights the potential to target modifiable characteristics in clusters for more targeted preventive intervention.

## Introduction

Stroke is a leading cause of death and disability globally with a substantial economic cost due to treatment and post-stroke care [1]. Patients at time of incident stroke have varied clinical characteristics, demographics, and biochemical profiles. This heterogeneity in characteristics at time of incident stroke impacts on cardiovascular morbidity and mortality outcomes [2]. Phenotyping (subgrouping) people after incident stroke, in terms of the risk of various cardiovascular outcomes, could provide individuals with the poorest prognosis better care. Intensive secondary prevention strategies including the use of novel medications such as proprotein convertase subtilisin/kexin type 9 (PCSK9) inhibitors and colchicine in patients at very high risk of adverse cardiovascular morbidity and mortality outcomes.

Cluster analysis, a hypothesis-free unsupervised machine learning data-driven approach, has been widely used to analyse clinical data to identify new phenotypic subgroups of complex and heterogeneous diseases including obstructive sleep apnoea [3], asthma [4,5], chronic obstructive pulmonary disease, chronic heart failure [6], dilated cardiomyopathy [7], sepsis [8], Parkinson's disease [9], breast cancer [10], and diabetes [11]. This approach does not include outcome data, and may be less biased in its results, especially when using retrospectively collected data. Clustering of clinical data may, therefore, be helpful in identifying subgroups of patients with incident stroke and generating new hypotheses. Efforts to determine such phenotypic groups in patients with incident stroke remain limited.

Using a large population-based cohort of adult patients with incident stroke, the objectives of this study are: (i) to identify patterns in linked primary and secondary clinical data and cluster patients based on phenotypic similarities; (ii) to assess the association between phenotypic clusters and subsequent recurrent stroke or CVD-related mortality, recurrent stroke or all-cause mortality, coronary heart disease (CHD), recurrent stroke, peripheral vascular disease (PVD), heart failure, CVD-related mortality, and all-cause mortality.

## Methods

### Study design and data source

This prospective population-based cohort study used the UK Clinical Practice Research Datalink (CPRD) GOLD database of anonymised longitudinal primary care electronic health records [12], linked to secondary care hospitalisation data (Hospital Episode Statistics [HES]) [13], national mortality data (Office for National Statistics [ONS]) [14], and social deprivation data (Index of Multiple Deprivation (IMD) 2015) [15]. Patients included in the CPRD GOLD database, from a network of general practices across the UK, are representative of the UK general population in terms of sex, age, and ethnicity [12].

### Study population

We identified a cohort of patients with incident non-fatal stroke in either primary care (CPRD GOLD) or secondary care (HES) between 1 January 1998 and 31 December 2017. Details about this cohort were previously reported [16]. Patients with a prior record of coronary heart disease (CHD), peripheral vascular disease (PVD), or heart failure before incident stroke event were excluded. Patients were followed from the date of incident stroke diagnosis until they developed a major adverse cardiovascular event (MACE), died, ceased contributing data, or last data collection date of the practice. The study flow diagram is shown in Fig 1.

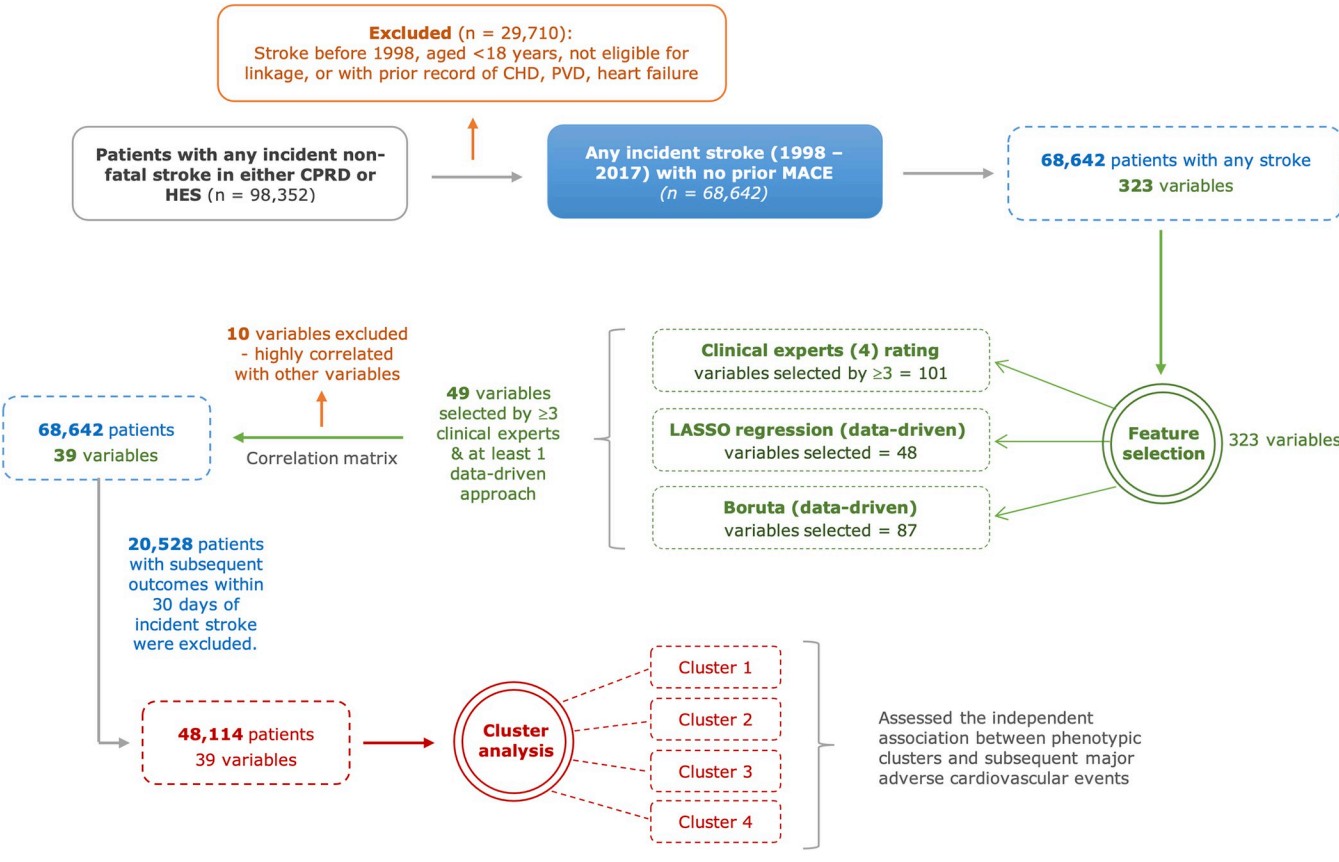

**Fig 1. Study flow diagram.**

## Outcomes

The primary outcome was a composite of recurrent stroke or CVD-related mortality event recorded after incident stroke from across the linked data sources (CPRD, HES or ONS registry). The secondary outcomes included: CHD, recurrent stroke, PVD, heart failure, CVD-related mortality, all-cause mortality, and the composite of recurrent stroke or all-cause mortality.

Subsequent outcomes within 30 days were considered to be representing or relating to the incident stroke event [16]. Analyses were, therefore, restricted to patients with subsequent outcomes occurring after 30 days of incident stroke.

## Potential candidate variables for phenotyping

Based on availability in the electronic health records and established association with CVD, 336 candidate variables were selected. These included demographic data, vital signs, biochemical parameters, comorbid conditions, and prescribed medications (S1 Table). For vital signs and biochemical test results, the most recent values/records within 24 months before incident stroke were extracted. A prescription within 12 months before incident stroke was considered as a medication prescribed. All comorbid conditions were defined based on the latest record of a comorbid condition any time before incident stroke. All code lists used have been published and available for download [17,18].

## Data processing

The variable distributions and missingness were first assessed. Multiple imputation by chained equations was used to account for missing data (S1 Fig, S2 Table). Ten imputed datasets were generated, using all available covariates and all the outcomes, although outcomes were not imputed [19,20]. The imputed datasets were pooled into a single dataset using Rubin's rules [21]. A high number of dimensions from a dataset with many variables/features is associated with a loss of meaningful differentiation between similar and dissimilar individuals–the 'curse of dimensionality' [22]. To improve the cluster analysis process and performance, feature selection was carried out to reduce collinearity, conditional dependence and noise contributing to increasing the variance. Feature selection was based on two (2) widely used data-driven feature selection methods (Boruta [23] and Least Absolute Shrinkage and Selection Operator (Lasso) regression [24]–S2 Fig) and clinical expert consensus. An expert group of clinicians from both primary (Consultant General Practitioners–NQ, JK) and secondary care (Stroke Medicine Consultant/Specialist–GN, GG) were independently consulted to attain consensus on which variables to select for the cluster analysis. Clinical expert consensus was defined as a 75% (3 out of 4) agreement among the clinical experts on each variable. 49 variables were rated important by the clinical experts and at least 1 of the 2 data-driven methods–S1 Table. After evaluating correlation among the 49 selected variables using mixedCor and Lares functions in R for mixed-type data (S3 Fig & S4 Fig), we excluded 10 highly correlated variables based on clinical judgement/importance. The remaining 39 variables, Box 1, were used for the cluster analysis.

## Phenotypic clustering

The prediction strength method by Tibshirani and Walther, 2015 [25] in the kamila function and the Elbow method were used to select the optimal number of clusters–S5 Fig. The *kamila* algorithm for mixed data clustering (S1 Text) was implemented to identify distinct patient phenotypic clusters. To ensure robustness of the clusters identified, 1,000 initialisations (that

### Box 1. Phenotypic domains and phenotypic variables used for cluster analysis

| Phenotypic domain | Phenotypic variables |
|---|---|
| Demographics | Age at incident stroke, sex, incident stroke sub-type, ethnicity, smoking status |
| Physical characteristics | Body mass index, diastolic and systolic blood pressures, pulse |
| Biochemical tests | C-reactive protein, glomerular filtration rate, haemoglobin, glycated haemoglobin, HDL cholesterol, LDL cholesterol, triglyceride |
| Comorbid conditions | Acute kidney injury, alcohol misuse, arrhythmia, cancer (composite), dementia, depression, diabetes mellitus (DM), DM with complications, diabetic ophthalmic complications, dyslipidaemia, hypertension, non-rheumatic aortic valve disorder, obesity, renal disease, severe mental illness, transient ischaemic attack |
| Prescribed medications | Anticoagulant, antidepressant, antidiabetic, antihypertensive, antiplatelet, diuretic, inotrope, loop diuretic, statin potency, thiazide diuretic |

is, random starting points) were carried out. Plot of the clusters with the principal component analysis (PCA) dimensions was generated (S6 Fig).

Using the h2o package (http://www.h2o.ai), a gradient boosting model was applied to identify as well as rank the key covariates (candidate variables) that predict each of the identified phenotypic clusters. The respective cluster groupings were coded as 1 –belonging to cluster or 0 –belonging to other clusters. SHAP (SHapley Additive exPlanations) was used to assess the discriminative influence of the variables for each of the identified clusters [26].

### Statistical analysis

For each cluster descriptive characteristics were provided, reporting proportion (%) for categorical variables and mean (SD) or median (IQR) for continuous variables. Kruskal-Wallis and chi-squared tests were used to compare across clusters, for continuous and categorical data, respectively.

The association between phenotypic clusters and adverse cardiovascular morbidity and mortality outcomes were assessed using Cox proportional hazards regression model. The hazard ratio (HR) for each phenotypic group is presented with 95% confidence intervals (CI) and corresponding $p$-values. Cumulative incidence plots were derived and differences between phenotypic groups assessed by the log-rank test. All statistical analyses were performed using Stata SE version 17 (StataCorp LP) and R version 4.1.0. An alpha level of 0.05 was used.

### Ethics approval and consent to participate

Ethical approval for this study was obtained from the Independent Scientific Advisory Committee (ISAC)–study protocol number 19_023R. De-identified (anonymised) patient data was obtained from the CPRD hence this study was exempt from obtaining informed consent from patients.

## Results

### Clinical characteristics among phenotypic clusters

We identified 68,642 patients aged $\geq$18 years old with any incident non-fatal stroke event between 1998 and 2017. A total of 20,528 (29.9%) patients with subsequent clinical outcomes

occurring within 30 days of incident stroke event were excluded, as these outcomes were considered to be related to the incident stroke event [16]. Cluster analysis was performed in the remaining 48,114 patients. Four phenotypic clusters with significant differences in clinical characteristics were identified. The identified clusters were numbered from 1 to 4 according to the ascendent overall incidence of subsequent composite outcome of recurrent stroke or CVD-related mortality, the primary outcome. Table 1 describes and compares the clinical characteristics among the phenotypic clusters.

The plots of the clusters are shown with the principal component analysis (PCA) dimensions in S6 Fig. The cluster profiles are summarised in Box 2.

### Box 2. Summary of cluster profiles

| Clusters | Number (%) | Characteristic feature(s) |
|---|---|---|
| Cluster 1 | 5,201 (10.8%) | Median age of 68 years (IQR 60–76), with a high proportion of patients who smoke or have diagnosed alcohol problems. Predominantly higher prevalence of CHD-related comorbidities/risk factors at time of incident stroke–high BMI (overweight/obese), diabetes, dyslipidaemia, hypertension, and family history of CVD. Higher proportion of antidiabetic and antihypertensive prescriptions. |
| Cluster 2 | 18,655 (38.8%) | Median age of 67 years (IQR 56–76), with lower prevalence of comorbid conditions at time of incident stroke. Higher proportion of smokers and patients with alcohol problems. Lowest proportion of prescribed medications. |
| Cluster 3 | 10,244 (21.3%) | Median age of 79 years (IQR: 73–85) with the highest prevalence of multiple long-term conditions at time of incident stroke–arrhythmia, cancer, chronic kidney disease, dementia, dyslipidaemia, hypertension, renal disease, and transient ischaemic attach. |
| Cluster 4 | 14,014 (29.1%) | The oldest cohort (median age: 83 years, IQR: 77–88) and predominantly female (75.4%). High prevalence of arrhythmia, dementia, and hypertension. |

### Variable importance for clusters

The supervised gradient boosting model to identify key covariates (candidate variables) that predict the respective phenotypic cluster had excellent prediction accuracy–area under the receiver operative curve (AUC) of 0.985, 0.982, 0.974, and 0.970 for clusters 1, 2, 3 and 4, respectively. The most common variables for predicting the respective phenotypic clusters were age at incident stroke, blood pressure, hypertension, LDL cholesterol, and potency of prescribed statin—Fig 2.

### Association with subsequent clinical outcomes

During the median follow-up time of 12.60 years (IQR, 7.60–16.97 years), there was a total of 24,588 (51.1%) composite recurrent stroke or CVD-related mortality outcome events. The occurrence of recurrent stroke + CVD-related mortality was different across the 4 phenotypic clusters–cluster 1 had the lowest incidence rate (15.13 per 100 person-years; 95% CI, 14.54–15.74), while cluster 4 had the highest incidence rate (23.17 per 100 person-years, 95% CI: 22.67–23.69). The risk of subsequent recurrent stroke + CVD-related mortality was significantly increased in cluster 2 (hazard ratio (HR), 1.07; 95% CI: 1.02–1.12); cluster 3 (HR, 1.20; 95% CI: 1.14–1.26), and cluster 4 (HR, 1.29; 95% CI: 1.26–1.33), when compared with cluster 1. Similar incidence rate and hazard ratio trends were observed for subsequent recurrent stroke + all-cause mortality outcome (cluster 2: HR, 1.07; 95% CI, 1.03–1.12; cluster 3: HR, 1.32, 95% CI: 1.26–1.37; cluster 4: HR, 1.54; 95% CI: 1.48–1.60) and recurrent stroke outcome

**Table 1. Characteristics of study population at time of incident stroke according to cluster membership (n = 48,114).**

| Characteristics | Entire cohort 48,114 (100%) | Cluster 1 5,201 (10.8%) | Cluster 2 18,655 (38.8%) | Cluster 3 10,244 (21.3%) | Cluster 4 14,014 (29.1%) |
|---|---|---|---|---|---|
| Follow-up in years, median (IQR) | 12.60 (7.60–16.97) | 13.63 (8.67–17.70) | 12.97 (7.97–17.26) | 13.74 (8.81–17.82) | 10.80 (6.02–15.53) |
| Females | 26,283 (54.6) | 2,120 (40.8) | 8,112 (43.5) | 5,490 (53.6) | 10,561 (75.4) |
| Age (years), mean (SD) | 76.0 (65.0–83.0) | 68.0 (60.0–76.0) | 67.0 (56.0–76.0) | 79.0 (73.0–85.0) | 83.0 (77.0–88.0) |
| Incident stroke subtype | | | | | |
| Haemorrhagic | 3,336 (6.9) | 216 (4.2) | 1,809 (9.7) | 484 (4.7) | 827 (5.9) |
| Ischaemic | 15,594 (32.4) | 1,896 (36.5) | 6,066 (32.5) | 2,797 (27.3) | 4,835 (34.5) |
| Stroke NOS | 29,184 (60.7) | 3,089 (59.4) | 10,780 (57.8) | 6,963 (68.0) | 8,352 (59.6) |
| Ethnicity | | | | | |
| Asian | 611 (1.3) | 157 (3.0) | 243 (1.3) | 150 (1.5) | 61 (0.4) |
| Black | 377 (0.8) | 87 (1.7) | 140 (0.8) | 69 (0.7) | 81 (0.6) |
| Mixed | 73 (0.2) | 12 (0.2) | 35 (0.2) | 13 (0.1) | 13 (0.1) |
| Other | 335 (0.7) | 50 (1.0) | 152 (0.8) | 66 (0.6) | 67 (0.5) |
| White | 43,011 (89.4) | 4,660 (89.6) | 16,589 (88.9) | 9,582 (93.5) | 12,180 (86.9) |
| Unknown | 3,707 (7.7) | 235 (4.5) | 1,496 (8.0) | 364 (3.6) | 1,612 (11.5) |
| Socioeconomic status | | | | | |
| 1 (Least deprived) | 10,292 (21.4) | 869 (16.7) | 3,849 (20.6) | 2,446 (23.9) | 3,128 (22.3) |
| 2 | 10,736 (22.3) | 1,056 (20.3) | 4,024 (21.6) | 2,426 (23.7) | 3,230 (23.0) |
| 3 | 10,355 (21.5) | 1,115 (21.4) | 4,004 (21.5) | 2,179 (21.3) | 3,057 (21.8) |
| 4 | 8,836 (18.4) | 1,066 (20.5) | 3,502 (18.8) | 1,744 (17.0) | 2,524 (18.0) |
| 5 (Most deprived) | 7,814 (16.2) | 1,093 (21.0) | 3,244 (17.4) | 1,438 (14.0) | 2,039 (14.5) |
| Unknown | 81 (0.2) | 2 (0.0) | 32 (0.2) | 11 (0.1) | 36 (0.3) |
| Current smokers | 8,357 (17.4) | 1,247 (24.0) | 4,791 (25.7) | 1,054 (10.3) | 1,265 (9.0) |
| Body mass index (kg/m$^2$) | 26.4 (25.0–28.0) | 30.0 (27.4–34.2) | 26.4 (25.2–27.6) | 25.8 (24.2–27.6) | 26.2 (25.0–27.4) |
| DBP (mmHg) | 80.0 (74.0–84.0) | 80.0 (76.0–89.0) | 80.0 (76.0–82.7) | 72.0 (68.0–80.0) | 80.0 (78.0–88.0) |
| SBP (mmHg) | 140.0 (130.0–148.0) | 142.0 (132.0–155.0) | 139.5 (130.0–144.0) | 133.0 (122.0–140.0) | 145.0 (139.6–160.0) |
| C-reactive protein | 9.8 (6.3–14.7) | 9.2 (6.0–14.8) | 10.1 (6.6–14.6) | 8.4 (5.3–13.9) | 10.4 (7.0–15.4) |
| Glomerular filtration rate | 67.2 (62.4–72.0) | 69.0 (61.2–75.0) | 68.0 (64.6–72.5) | 65.3 (58.0–72.0) | 66.4 (61.8–70.4) |
| Glycated haemoglobin | 49.9 (46.7–53.4) | 58.3 (53.0–66.4) | 49.7 (47.0–52.4) | 47.5 (44.3–51.0) | 50.2 (47.4–53.3) |
| Haemoglobin | 13.5 (12.9–14.2) | 14.2 (13.3–15.0) | 13.6 (13.2–14.3) | 13.2 (12.3–14.1) | 13.4 (12.7–13.9) |
| HDL cholesterol (mmol/L) | 1.5 (1.3–1.6) | 1.2 (1.0–1.3) | 1.5 (1.3–1.6) | 1.5 (1.3–1.7) | 1.5 (1.4–1.7) |
| LDL cholesterol (mmol/L) | 3.0 (2.6–3.3) | 3.0 (2.3–3.5) | 3.0 (2.8–3.3) | 2.4 (1.9–2.8) | 3.1 (2.9–3.4) |
| Total cholesterol (mmol/L) | 5.1 (4.7–5.4) | 5.1 (4.3–5.8) | 5.1 (4.8–5.4) | 4.5 (3.9–4.9) | 5.3 (5.0–5.7) |
| Triglyceride (mmol/L) | 1.4 (1.2–1.7) | 2.1 (1.6–2.7) | 1.4 (1.3–1.6) | 1.2 (1.0–1.4) | 1.5 (1.3–1.6) |
| Pulse | 76.4 (73.9–79.0) | 77.8 (74.9–80.8) | 76.6 (74.4–78.9) | 74.8 (71.8–77.7) | 76.7 (74.4–79.3) |
| Acute kidney injury | 218 (0.5) | 47 (0.9) | 44 (0.2) | 84 (0.8) | 43 (0.3) |
| Alcohol problem | 1,345 (2.8) | 217 (4.2) | 779 (4.2) | 221 (2.2) | 128 (0.9) |
| Arrhythmia | 4,575 (9.5) | 362 (7.0) | 569 (3.1) | 1,955 (19.1) | 1,689 (12.1) |
| Atrial fibrillation | 4,210 (8.8) | 325 (6.3) | 496 (2.7) | 1,838 (17.9) | 1,551 (11.1) |
| Cancer | 7,652 (15.9) | 634 (12.2) | 2,167 (11.6) | 2,514 (24.5) | 2,337 (16.7) |
| Chronic kidney disease | 4,945 (10.3) | 767 (14.8) | 390 (2.1) | 2,580 (25.2) | 1,208 (8.6) |
| Dementia | 2,489 (5.2) | 80 (1.5) | 647 (3.5) | 775 (7.6) | 987 (7.0) |
| Depression | 9,147 (19.0) | 1,327 (25.5) | 3,589 (19.2) | 1,800 (17.6) | 2,431 (17.3) |
| Diabetes mellitus | 5,494 (11.4) | 2,702 (52.0) | 392 (2.1) | 1,985 (19.4) | 415 (3.0) |
| Dyslipidaemia | 4,845 (10.1) | 1,154 (22.2) | 927 (5.0) | 2,128 (20.8) | 636 (4.5) |
| Family history of CVD | 8,817 (18.3) | 1,240 (23.8) | 3,229 (17.3) | 2,278 (22.2) | 2,070 (14.8) |
| Hypertension | 22,447 (46.7) | 3820 (73.4) | 1723 (9.2) | 7885 (77.0) | 9019 (64.4) |

*(Continued)*

**Table 1.** (Continued)

| Characteristics | Entire cohort 48,114 (100%) | Cluster 1 5,201 (10.8%) | Cluster 2 18,655 (38.8%) | Cluster 3 10,244 (21.3%) | Cluster 4 14,014 (29.1%) |
|---|---|---|---|---|---|
| Non-rheumatic aortic valve disorder | 571 (1.2) | 46 (0.9) | 74 (0.4) | 254 (2.5) | 197 (1.4) |
| Renal disease | 5,545 (11.5) | 867 (16.7) | 555 (3.0) | 2,764 (27.0) | 1,359 (9.7) |
| Severe mental illness | 695 (1.4) | 108 (2.1) | 327 (1.8) | 102 (1.0) | 158 (1.1) |
| Transient ischaemic attack | 12,373 (25.7) | 1,326 (25.5) | 3,345 (17.9) | 4,881 (47.6) | 2,821 (20.1) |
| Anti-arrhythmic | 2,163 (4.5) | 227 (4.4) | 451 (2.4) | 698 (6.8) | 787 (5.6) |
| Anti-coagulant | 2,807 (5.8) | 286 (5.5) | 486 (2.6) | 1,225 (12.0) | 810 (5.8) |
| Anti-depressant | 11,212 (23.3) | 1,508 (29.0) | 3,965 (21.3) | 2,412 (23.5) | 3,327 (23.7) |
| Anti-diabetics | 4,379 (9.1) | 2,476 (47.6) | 254 (1.4) | 1,421 (13.9) | 228 (1.6) |
| Anti-hypertensive | 23,678 (49.2) | 4,231 (81.3) | 2,312 (12.4) | 8,497 (82.9) | 8,638 (61.6) |
| Anti-platelet | 19,789 (41.1) | 2,618 (50.3) | 4,605 (24.7) | 6,753 (65.9) | 5,813 (41.5) |
| Diuretics | 16,835 (35.0) | 2,265 (43.5) | 280 (1.5) | 5,288 (51.6) | 9,002 (64.2) |
| Inotropic | 2,084 (4.3) | 141 (2.7) | 160 (0.9) | 714 (7.0) | 1,069 (7.6) |
| Statin | | | | | |
| Low intensity | 1,855 (3.9) | 391 (7.5) | 321 (1.7) | 860 (8.4) | 283 (2.0) |
| Moderate intensity | 9,797 (20.4) | 1,939 (37.3) | 1,889 (10.1) | 5,177 (50.5) | 792 (5.7) |
| High intensity | 2,240 (4.7) | 713 (13.7) | 335 (1.8) | 1,062 (10.4) | 130 (0.9) |

CVD: cardiovascular disease; DBP: diastolic blood pressure; HDL: high density lipoprotein; LDL: low density lipoprotein; n: frequency/numbers; SBP: systolic blood pressure; SD: standard deviation; %: percent.

(cluster 2: HR, 1.10; 95% CI, 1.05–1.16; cluster 3: HR, 1.12, 95% CI, 1.06–1.18; cluster 4: HR, 1.25; 95% CI: 1.19–1.32).

Different trends in incidence rate and hazard ratios were observed, however, for subsequent CHD, PVD, heart failure, CVD-related and all-cause mortality outcomes–Fig 3 and Table 2. When compared with cluster 1, the risk of subsequent CHD events was significantly decreased in the other 3 clusters (cluster 2: HR, 0.49; 95% CI: 0.44–0.55; cluster 3: HR, 0.64; 95% CI, 0.56–0.73; cluster 4: HR, 0.55; 95% CI, 0.49–0.63). A similar decreased risk in the other 3 clusters when compared to cluster 1 was observed for risk of subsequent PVD.

For risk of subsequent heart failure, CVD-related mortality and all-cause mortality, cluster 2 had a significantly decreased risk when compared to cluster 1 while clusters 3 and 4 had a significantly increased risk–Table 2. The occurrence of subsequent cardiovascular morbidity and mortality outcomes across the different phenotypic clusters is presented as Kaplan Meier plots in Fig 4.

## Discussion

This population-based study exploring phenotypic characteristics of patients with incident stroke using a data-driven-cluster analysis approach identified four clinically meaningful patient clusters based on the phenotypic characteristics at time of incident stroke. There was a varied relationship between the identified phenotypic clusters and subsequent risk of adverse cardiovascular morbidity and mortality outcomes.

In our study, four distinct and clinically meaningful phenotypic clusters were identified. Smoking, a strong independent modifiable risk factor for cardiovascular morbidity and mortality outcomes [27], was most highly prevalent in clusters 1 and 2. Preventative strategy to communicate the risks of smoking and the benefits of quitting to this cluster of patients could be an effective means to promote smoking cessation and reduce risk for subsequent adverse

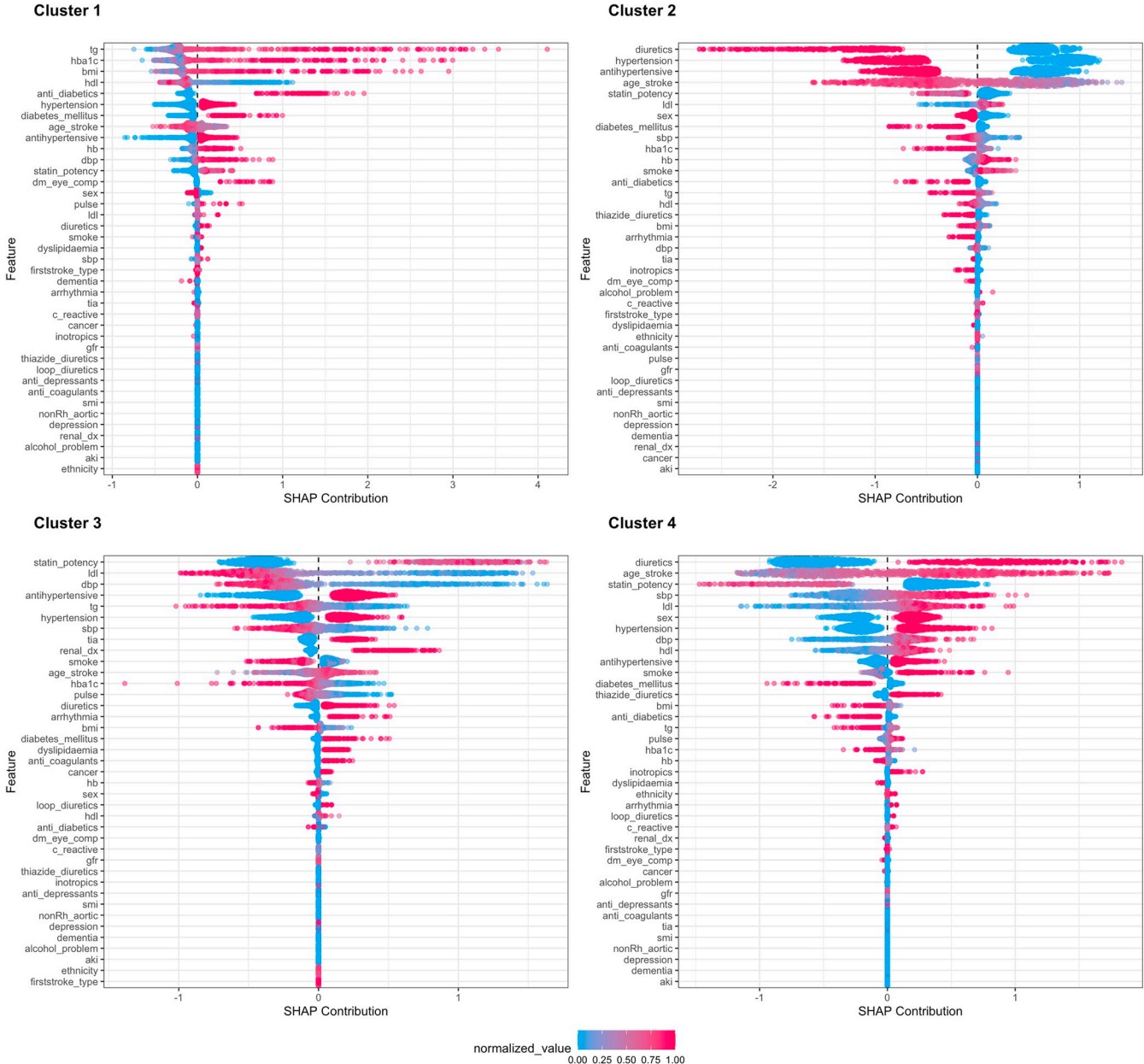

**Fig 2. Plot showing the clinical parameters which are the core of each phenotypic cluster.** aki: acute kidney injury; dbp: diastolic blood pressure; dm_eye_comp: diabetic ophthalmic complications; sbp: systolic blood pressure; gfr: glomerular filtration rate; hb: haemoglobin; hdl: high-density lipoprotein cholesterol; ldl: low-density lipoprotein cholesterol; hba1c: glycated haemoglobin; nonRH_aortic: non-rheumatic aortic valve disorder; smi: severe mental illness; tg: triglyceride; tia: transient ischaemic attack. SHAP summary plot combines feature/variable importance with feature effects. Each point on the summary plot is a Shapley value for an individual. The position on the y-axis is determined by the feature and on the x-axis by the Shapley value. The colour represents the value from low to high. The features are ordered according to importance.

events [28]. With the exception of clusters 2, the 3 other clusters included had high prevalence of multiple long-term conditions as well as CVD risk factors at time of incident stroke. Patients with incident stroke have been shown to commonly have pre-existing long-term conditions [29]. To optimally manage the possible atherogenic effect of these comorbid condition to reduce risk of subsequent cardiovascular morbidity and mortality outcomes, both non-pharmacological (that is, lifestyle modification [30,31]) and pharmacological (antihypertensives for

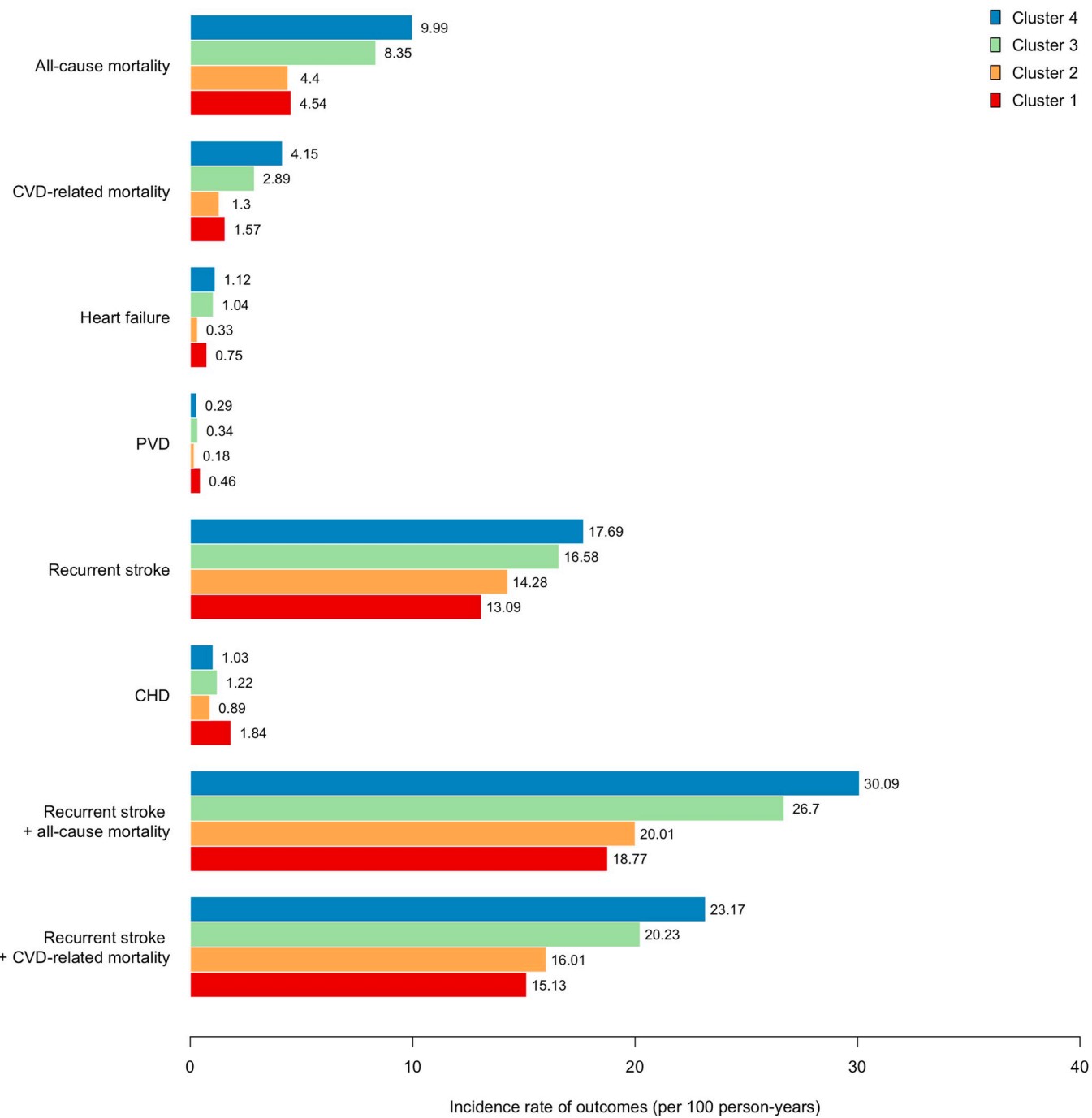

**Fig 3. Incidence rate for the subsequent adverse outcomes by the identified phenotypic clusters.**

blood pressure management [32]; lipid-lowering medications such as statins for cholesterol management [33]; antidiabetics for blood sugar control [30]; and antiplatelets/anticoagulants to manage arrhythmia [34]) strategies need to be prioritised in line with clinical guidelines [35]. Frequent monitoring/reviews to ensure treatment targets are being met is important [36]. Age, a non-modifiable risk factor, was a key factor for the patient cluster membership. Among older adults (typical of cluster 4), incidence of aortic disease, PVD and venous

**Table 2. Subsequent major adverse outcomes after incident stroke by phenotypic clusters.**

|  | | Events Number | Incidence rate (95% CI) per 100 PY | Hazard ratio (95% CI) |
|---|---|---|---|---|
| Recurrent stroke + CVD-related mortality | | 24,588 | 18.53 (18.30–18.76) | |
| | Cluster 1 | 2,447 | 15.13 (14.54–15.74) | Reference |
| | Cluster 2 | 9,249 | 16.01 (15.68–16.33) | 1.07 (1.02–1.12) |
| | Cluster 3 | 4,980 | 20.23 (19.68–20.80) | 1.20 (1.14–1.26) |
| | Cluster 4 | 7,912 | 23.17 (22.67–23.69) | 1.44 (1.37–1.50) |
| Recurrent stroke + all-cause mortality | | 33,891 | 23.78 (23.52–24.03) | |
| | Cluster 1 | 3,183 | 18.77 (18.13–19.43) | Reference |
| | Cluster 2 | 12,275 | 20.01 (19.66–20.37) | 1.07 (1.03–1.12) |
| | Cluster 3 | 7,121 | 26.70 (26.09–27.33) | 1.32 (1.26–1.37) |
| | Cluster 4 | 11,312 | 30.09 (29.54–30.65) | 1.54 (1.48–1.60) |
| Coronary heart disease (All) | | 2119 | 1.09 (1.04–1.14) | |
| | Cluster 1 | 408 | 1.84 (1.67–2.02) | Reference |
| | Cluster 2 | 784 | 0.89 (0.83–0.95) | 0.49 (0.44–0.55) |
| | Cluster 3 | 419 | 1.22 (1.10–1.34) | 0.64 (0.56–0.73) |
| | Cluster 4 | 508 | 1.03 (0.94–1.12) | 0.55 (0.49–0.63) |
| Recurrent stroke (All) | | 19,810 | 15.42 (15.21–15.63) | |
| | Cluster 1 | 2,075 | 13.09 (12.54–13.67) | Reference |
| | Cluster 2 | 8,053 | 14.28 (13.97–14.59) | 1.10 (1.05–1.16) |
| | Cluster 3 | 3,939 | 16.58 (16.07–17.11) | 1.12 (1.06–1.18) |
| | Cluster 4 | 5,743 | 17.69 (17.24–18.15) | 1.25 (1.19–1.32) |
| Peripheral arterial disease (All) | | 529 | 0.27 (0.24–0.29) | |
| | Cluster 1 | 105 | 0.46 (0.38–0.55) | Reference |
| | Cluster 2 | 161 | 0.18 (0.15–0.21) | 0.40 (0.31–0.51) |
| | Cluster 3 | 118 | 0.34 (0.28–0.40) | 0.70 (0.54–0.91) |
| | Cluster 4 | 145 | 0.29 (0.24–0.34) | 0.62 (0.48–0.79) |
| Heart failure (All) | | 1390 | 0.70 (0.67–0.74) | |
| | Cluster 1 | 172 | 0.75 (0.64–0.87) | Reference |
| | Cluster 2 | 295 | 0.33 (0.29–0.37) | 0.44 (0.37–0.53) |
| | Cluster 3 | 363 | 1.04 (0.94–1.15) | 1.34 (1.12–1.61) |
| | Cluster 4 | 560 | 1.12 (1.03–1.22) | 1.48 (1.24–1.75) |
| Cardiovascular mortality (All) | | 4,778 | 2.34 (2.27–2.41) | |
| | Cluster 1 | 372 | 1.57 (1.42–1.74) | Reference |
| | Cluster 2 | 1,196 | 1.30 (1.22–1.37) | 0.85 (0.75–0.95) |
| | Cluster 3 | 1,041 | 2.89 (2.72–3.07) | 1.69 (1.50–1.91) |
| | Cluster 4 | 2,169 | 4.15 (3.98–4.33) | 2.52 (2.25–2.81) |
| All-cause mortality (All) | | 14,081 | 6.58 (6.47–6.68) | |
| | Cluster 1 | 1,108 | 4.54 (4.28–4.81) | Reference |
| | Cluster 2 | 4,222 | 4.40 (4.27–4.54) | 0.98 (0.92–1.05) |
| | Cluster 3 | 3,182 | 8.35 (8.06–8.64) | 1.76 (1.64–1.88) |
| | Cluster 4 | 5,569 | 9.99 (9.73–10.26) | 2.14 (2.01–2.29) |

CI: confidence interval; PY: person year

thromboembolism increase as age-related alterations in vascular structure and function are compounded by the longer exposure to CVD risk factors [37].

Clustering is a common approach used to analyse large datasets, to identify both the number of subgroups in the data and the attributes of each subgroup, as has been done in this

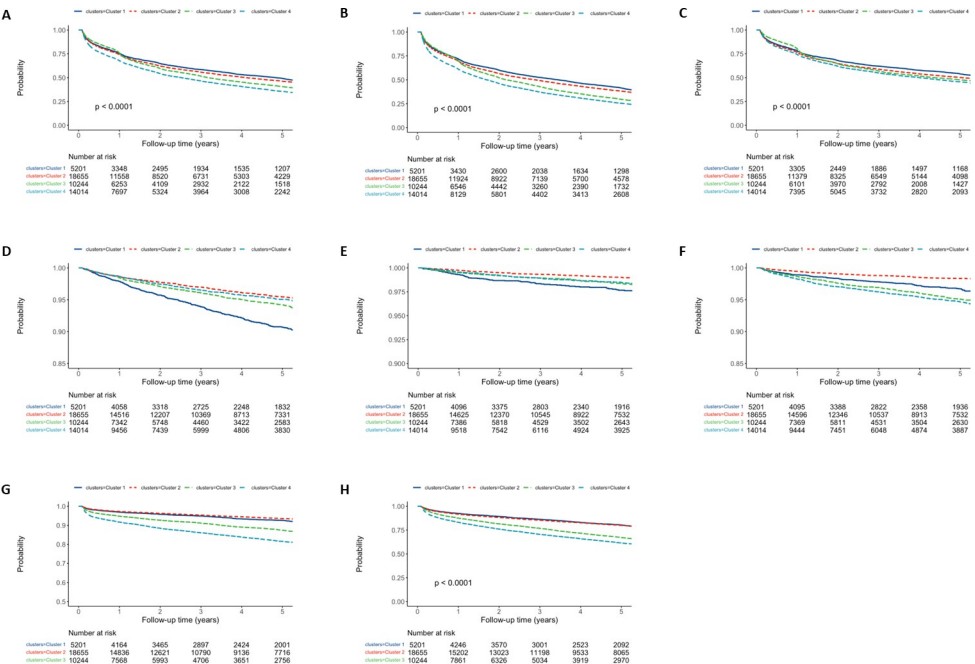

**Fig 4. Kaplan-Meier plots for subsequent clinical outcomes stratified by phenotypic clusters. A**: Recurrent stroke and CVD-related mortality (log-rank p<0.0001); **B**: Recurrent stroke and all-cause mortality (log-rank p<0.0001); **C**: Recurrent stroke (log-rank p<0.0001); **D**: Coronary heart disease (log-rank p<0.0001); **E**: Peripheral vascular disease (log-rank p<0.0001); **F**: Heart failure (log-rank p<0.0001); **G**: Cardiovascular-related mortality (log-rank p<0.0001); **H**: All-cause mortality (log-rank p<0.0001).

study. Data analysed in real applications including healthcare (from electronic health records) are mostly characterised by a mix of continuous and categorial variables. More common approaches that have been applied to mixed data include converting the variables to a single data type by either coding the categorical variables as numbers or dummy coding the variables and then applying standard distance methods such as k-means designed for continuous variables to the transformed data to achieve the clustering objective(s) [38,39]. Continuous variables have also been converted to categorical variables using interval-based bucketing [40,41]. Similarities that may have been observed in the original data may be lost when the data is transformed in such ways [40]. Kamila clustering algorithm has, however, been shown to better handle high imbalance between continuous and categorical data than any other method [40,42]. From a computational perspective, when compared with other algorithms, the Kamila algorithm offers the best performance and most time-efficient when dealing with large datasets (in relation to both observations and variables) in the setting of heterogeneous data, as was the situation in our study [40,42].

## Strengths and limitations

To our knowledge, this is the first time that a data-driven cluster analysis aimed at identifying stroke phenotypes in a well characterised large population-based cohort of adults with any incident stroke. This allows us to cover a large range of stroke phenotypes. Most importantly, we had a comprehensive linked database with a broad spectrum of clinical data with many of these variables being explored in cluster analysis for the first time.

There are, however, limitations of this study worth considering. First and foremost, the study was not meant to propose a new classification for stroke, because the clusters are likely

to vary according to patient characteristics and available data. These results serve to underscore the need for novel multidimensional stroke classification approaches for improving patient care. Furthermore, they are aimed to generate hypotheses for future studies that will integrate clinical and biological data in patients, with the goal of improving the care of patients with stroke. With immense advancement in machine learning, cluster analysis can be performed in a large number of ways [42,43]. However, the knowledge and experience of the relevant experts remain the best judge in the interpretation of findings from cluster analysis, hence the involvement of a diverse group of clinical specialists, clinical researchers, and data experts in our study. The presence of missing data is a common occurrence in clinical research using electronic health records collected as part of routine care. For example, laboratory tests are typically requested only when considered necessary for a patient's health condition. Similarly, information on BMI or smoking status may not be consistently recorded, leading to potential bias in patterns of data completeness. To address this issue, multiple imputation by chained equations, as outlined in the methods section, was used to handle missing data in our study, which is the preferred option under any missingness mechanism [19,20].

## Implications

Cluster analysis is most suited to address the multidimensional complexity of disease conditions with considerable heterogeneity such as stroke. Population-based cluster analysis could provide further understanding of disease patterns. Additionally, patients could be phenotyped and allocated to specific clusters that could be associated with different risks for various outcomes. Different treatment strategies or interventions could be targeted at specific phenotypic clusters, based available evidence on risk and possible response. Future clinic trial design could also focus on high-risk clusters or focus on specific aspects within a cluster.

## Conclusions

Using an unsupervised learning data-driven cluster analysis on a broad spectrum of baseline clinical data of patients with incident stroke, we identified four phenotypic and clinically meaningful clusters with respect to risk of subsequent major adverse outcomes. These findings highlight the significant heterogeneity that exists within patients with incident stroke with respect to subsequent adverse outcomes. This offers an opportunity to revisit the stratification of care for patients with incident stroke to improve patient outcomes. Further exploration in different patient cohorts and populations is needed.

## Supporting information

**S1 Text. Additional Methods.**
(DOCX)

**S1 Fig. All clinical variables with missing values.**
(DOCX)

**S2 Fig. Feature selection.**
(DOCX)

**S3 Fig. Plot of correlation matrix of 49 selected variables.**
(DOCX)

**S4 Fig. Ranked cross-correlation plot of 49 selected variables.**
(DOCX)

**S5 Fig. Optimal number of clusters.**
(DOCX)

**S6 Fig. Principal component analysis (PCA) plots.**
(DOCX)

**S1 Table. Overview of all variables and the in- or exclusion at the various data processing steps.**
(DOCX)

**S2 Table. Observed versus imputed values after multiple imputation for all clinical variables with missing data.**
(DOCX)

## Acknowledgments

We thank the practices that contributed to the CPRD GOLD.

## Author Contributions

**Conceptualization:** Ralph K. Akyea, Stephen F. Weng, Nadeem Qureshi.

**Data curation:** Ralph K. Akyea.

**Formal analysis:** Ralph K. Akyea.

**Funding acquisition:** Ralph K. Akyea, Joe Kai, Stephen F. Weng, Nadeem Qureshi.

**Investigation:** Ralph K. Akyea.

**Methodology:** Ralph K. Akyea, Stephen F. Weng.

**Resources:** Ralph K. Akyea.

**Supervision:** Stephen F. Weng, Nadeem Qureshi.

**Visualization:** Ralph K. Akyea.

**Writing – original draft:** Ralph K. Akyea.

**Writing – review & editing:** Ralph K. Akyea, George Ntaios, Evangelos Kontopantelis, Georgios Georgiopoulos, Daniele Soria, Folkert W. Asselbergs, Joe Kai, Stephen F. Weng, Nadeem Qureshi.

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
