## [Decision Letter · Decision Letter 0]

7 Jun 2023

PDIG-D-23-00084

A population-based study exploring phenotypic clusters and clinical outcomes in stroke using unsupervised machine learning approach

PLOS Digital Health

Dear Dr. Akyea,

Thank you for submitting your manuscript to PLOS Digital Health. After careful consideration, we feel that it has merit but does not fully meet PLOS Digital Health's publication criteria as it currently stands. Therefore, we invite you to submit a revised version of the manuscript that addresses the points raised during the review process.

Please submit your revised manuscript within 30 days Jul 07 2023 11:59PM. If you will need more time than this to complete your revisions, please reply to this message or contact the journal office at digitalhealth@plos.org. Please include the following items when submitting your revised manuscript:

We look forward to receiving your revised manuscript.

Kind regards,

Gilles Guillot

Academic Editor

PLOS Digital Health

Journal Requirements:

Reviewers' comments:

Reviewer's Responses to Questions

**Comments to the Author**

1. Does this manuscript meet PLOS Digital Health’s publication criteria? Is the manuscript technically sound, and do the data support the conclusions? The manuscript must describe methodologically and ethically rigorous research with conclusions that are appropriately drawn based on the data presented.

Reviewer #1: Yes

Reviewer #2: Yes

2. Has the statistical analysis been performed appropriately and rigorously?

Reviewer #1: Yes

Reviewer #2: Yes

3. Have the authors made all data underlying the findings in their manuscript fully available (please refer to the Data Availability Statement at the start of the manuscript PDF file)?

Reviewer #1: Yes

Reviewer #2: Yes

4. Is the manuscript presented in an intelligible fashion and written in standard English?

Reviewer #1: Yes

Reviewer #2: Yes

5. Review Comments to the Author

Reviewer #1: The study is well designed and the manuscript is well written. The dataset and preprocessing steps are introduced carefully. Method and interpretable method is very benificial for the research domain.

Reviewer #2: This study offers a data-driven approach to stratify individuals with incident stroke into phenotypic clusters and identify the risk of major adverse outcomes. This is a very interesting approach that can help redefine patients risks. However, I have a few remarks:

Why did you choose a composite outcome for your primary outcome? Is this based on recommendations by clinicians? If so, you should explicit this choice in the methodology, as composite criteria are never ideal for this type of studies. However, if this choice is based on real-life experience (and thus, justified), you should explain it in the manuscript.

While multiple imputation is the correct approach for missing data, I would like to see more info about this data from a clinician point of view: is this data missing because it’s not relevant for these type of patients (and maybe this variable should not be included in the model) or are the patients with missing data different from other patients? Since you are trying to identify patients with phenotypic similarities, the reason why this data is missing could be relevant.

Why did you use chi-squared tests for categorical data? Why not Fisher tests? You used a non-parametric test for continuous data and a parametric test for categorical data. I think you should either use non-parametric tests for both type of data or justify your methodological choice.

It’s interesting to see that ethnicity does not seem to influence the outcome, even though previous studies have shown links between characteristics such as BMI or hypertension and ethnicity (higher hypertension amongst black people, cardiovascular risk with lower BMI in people of East Asian descent...) However, the number of non-white patients is low in the database, which could explain this result. It would be great to compare these results with databases with a majority of black patients or a majority of Asian patients to confirm that ethnicity does not have a big influence on the risk of subsequent major adverse outcomes. Unfortunately, I’m not sure this is feasible at this moment.

6. PLOS authors have the option to publish the peer review history of their article (what does this mean?). If published, this will include your full peer review and any attached files.

**Do you want your identity to be public for this peer review?** For information about this choice, including consent withdrawal, please see our Privacy Policy.

Reviewer #1: No

Reviewer #2: No

---

## [Editor Report · Decision Letter 1]

19 Jul 2023

A population-based study exploring phenotypic clusters and clinical outcomes in stroke using unsupervised machine learning approach

PDIG-D-23-00084R1

Dear Dr Akyea,

We are pleased to inform you that your manuscript 'A population-based study exploring phenotypic clusters and clinical outcomes in stroke using unsupervised machine learning approach' has been provisionally accepted for publication in PLOS Digital Health.

Best regards,

Gilles Guillot

Academic Editor

PLOS Digital Health